# Functions of Vertebrate Ferlins

**DOI:** 10.3390/cells9030534

**Published:** 2020-02-25

**Authors:** Anna V. Bulankina, Sven Thoms

**Affiliations:** 1Department of Internal Medicine 1, Goethe University Hospital Frankfurt, 60590 Frankfurt, Germany; Anna.Bulankina@kgu.de; 2Department of Child and Adolescent Health, University Medical Center Göttingen, 37075 Göttingen, Germany

**Keywords:** dysferlin, myoferlin, otoferlin, C2 domain, calcium-sensor, muscular dystrophy, dysferlinopathy, limb girdle muscular dystrophy type 2B (LGMD2B), membrane repair, T-tubule system, DFNB9

## Abstract

Ferlins are multiple-C2-domain proteins involved in Ca^2+^-triggered membrane dynamics within the secretory, endocytic and lysosomal pathways. In bony vertebrates there are six ferlin genes encoding, in humans, dysferlin, otoferlin, myoferlin, Fer1L5 and 6 and the long noncoding RNA Fer1L4. Mutations in *DYSF* (dysferlin) can cause a range of muscle diseases with various clinical manifestations collectively known as dysferlinopathies, including limb-girdle muscular dystrophy type 2B (LGMD2B) and Miyoshi myopathy. A mutation in *MYOF* (myoferlin) was linked to a muscular dystrophy accompanied by cardiomyopathy. Mutations in *OTOF* (otoferlin) can be the cause of nonsyndromic deafness DFNB9. Dysregulated expression of any human ferlin may be associated with development of cancer. This review provides a detailed description of functions of the vertebrate ferlins with a focus on muscle ferlins and discusses the mechanisms leading to disease development.

## 1. Introduction

Ferlins belong to the superfamily of proteins with multiple C2 domains (MC2D) that share common functions in tethering membrane-bound organelles or recruiting proteins to cellular membranes. Ferlins are described as calcium ions (Ca^2+^)-sensors for vesicular trafficking capable of sculpturing membranes [1,2,3]. Ferlins of bony vertebrates (humans and the model organisms zebrafish and mice) are among the largest proteins in this superfamily with molecular weights of more than 200 kDa. Their hallmark is the presence of five to seven C2 domains in the cytoplasmic segment and a single transmembrane domain near the C-terminus, defining them as tail-anchored proteins. Phylogenetic analysis of ferlins within bony vertebrates shows variability in both the presence/absence of individual members of the ferlin family and the numbers of predicted C2 domains within the subgroups. Proper function of ferlins, in particular dysferlin, myoferlin and otoferlin is important for human health [4,5,6,7,8].

In this review, we present a summary on ferlins structure and function in health and disease. We focus the discussion on dysferlin, the most-studied ferlin protein. For more complete and complementary information, the reader is directed to excellent review articles that have been published in the last decade [8,9,10,11].

## 2. Proteins with Multiple C2 Domains (MC2D)

The superfamily of MC2D containing proteins includes members with two to seven confirmed or predicted C2 domains [12] (Figure 1). A C2 domain consists of 100–130 amino acids, often binds Ca^2+^ and negatively charged lipids like phosphatidylserine (PS) or phosphatidylinositol 4,5-bisphosphate (PIP2), thereby mediating interaction with membranes. The proteins of this superfamily act as Ca^2+^-sensors and organizers of vesicular trafficking, signaling, lipid transfer and as enzymes for lipid modification. To fulfill these functions, MC2D proteins tether membranous organelles or recruit proteins to membranes. Interestingly, members of ten out of approximately twelve protein families within this superfamily function presynaptically. The scaffolding proteins Piccolo, RIM1 and RIM2 (Ras-related in brain 3 (Rab3)-interacting molecules) participate in the organization of the presynaptic active zone and recruitment of synaptic vesicles (SVs) to the membrane [13,14,15]. Munc13-1 and -2 regulate SVs docking to the active zone and their priming for exocytosis [16,17]. The ability to dock SVs and to promote membrane fusion as Ca^2+^-sensors is well described for synaptotagmins I, II and VII [18,19,20,21]. DOC2B also acts as a Ca^2+^-sensor in SV exocytosis, while both, DOC2B and Rabphilin promote priming of SVs [21]. Apart from that, copine-6 acts as a suppressor of spontaneous neurotransmission [22], whereas two presynaptic ER-resident proteins in *Drosophila*, extended synaptotagmin [23] and multiple C2 and transmembrane domain protein (MCTP) [24] promote neurotransmission. The ferlin protein family is no exception in this case and otoferlin is mandatory for SV exocytosis at the first auditory synapse in the mammalian cochlea and in sensory hair cells of zebrafish [25,26].

In addition to controlling presynaptic function in neurons and vesicle or organelle exocytosis in non-neuronal cells, MC2D proteins:Mediate tethering of the ER to the plasma membrane (PM) and lipid transfer (extended synaptotagmins) [27],Recruit proteins to membranes and are involved in cell signaling (copines and RASAL1) [28,29,30], and,Phosphorylate inositol phospholipids, thereby influencing intracellular processes like signal transduction or clathrin-mediated endocytosis (PI3KC2s) [31].

Thus, MC2D proteins perform diverse functions and many of them control vesicular trafficking, in particular, neurotransmitter release, in one or the other way.

## 3. Vertebrate Ferlins: Family Members and Domain Organization

In the following discussion, we focused on the vertebrates—zebrafish (*Danio rerio*), mice (*Mus musculus*) and humans (*Homo sapiens*)—representing important (model) organisms for the study of ferlin functions. Six ferlin genes were present in each of these organisms (Figure 2). The phylogenetic analysis of the corresponding proteins demonstrating the evolutionary relationship between ferlins is shown in Figure 3. 

Human ferlin genes include five protein-encoding members, *FER1L1/DYSF* (dysferlin), *FER1L2/OTOF* (otoferlin), *FER1L3/MYOF* (myoferlin), *FER1L5* (Fer1L5) and *FER1L6* (Fer1L6), and the pseudogene *FER1L4* encoding a long non-coding RNA [9,32,33]. The full set of six ferlin proteins (Fer1l1–6) is expressed in the mouse only. The zebrafish genome contains two otoferlin genes, *otofa* and *otofb,* on different chromosomes [26], while no *fer1l5* ortholog appears to be present [34] (Figure 2 and Figure 3). It is tempting to speculate that the duplication of the otoferlin gene in zebrafish may parallel the development of the lateral line and inner ear, and it is not present in higher vertebrates that have also no lateral line, since both otoferlin a and b are expressed in the otic placode (giving rise to the inner ear), but only otoferlin b transcripts were detected in the lateral line [26].

C2 domain organization of vertebrate ferlins shows variability in the number of predicted domains and the C2 domain layout is conserved in Fer1L5s only (Figure 2). All other subgroups (dysferlins, otoferlins, myoferlins, Fer1L4s and Fer1L6s) show one outlier each, which has lost or gained one C2 domain. In addition to the C2 domains, ferlins of the bony vertebrates possess all or some of the specific homology domains, namely FerI, FerA, FerB and the ‘embedded’ DysF domain (Figure 1 and Figure 2). Dysferlin, myoferlin and Fer1L5 contain all of these homology domains. On the basis of the presence of the embedded DysF domain, they are collectively known as type I ferlins. In contrast, otoferlin, Fer1l4 and Fer1L6 lack DysF domains and thus represent type II ferlins [9]. Of note, FerA domains are apparently not conserved in the primary sequence of type II ferlins of bony vertebrates, however, the characteristic to type I ferlins four amphipathic helix bundle fold of FerA domain is present in human otoferlin and such structural element can be predicted in all ferlin proteins [35]. In summary, the most conserved ferlin domains are the C2B-FerI-C2C stretch and the FerB, C2D and C2F domains as summarized in Figure 1 and Figure 2.

## 4. Ferlin Domains: Properties and Function

Ca^2+^- and PS-binding properties of individual C2 domains of human dysferlin and otoferlin were characterized, and all seven dysferlin and five of six otoferlin C2 domains bind Ca^2+^ and PS-containing liposomes [40,41,42,43]. The binding of the C2 domains and of truncated dysferlin, otoferlin and myoferlin constructs changes the packaging of PS-containing bilayers in vitro, bearing the potential to sculpture the membranes in vivo [44]. In addition to their Ca^2+^- and PS-binding properties, two of the six otoferlin C2 domains are known to interact with PIP2 [40]. The dysferlin C2A domain also binds PIP2 and phosphatidylinositol 4-phosphate [45]. C2 domains are also reported to participate in protein-protein interactions and mediate the dimerization of dysferlin [46]. Dysferlin, myoferlin and otoferlin FerA domains are capable of binding to phospholipid membranes and this interaction is enhanced by the presence of Ca^2+^ [35]. Interestingly, one of the most conserved ferlin segments, C2B-FerI-C2C, regulates dysferlin PM expression and rate of its endocytosis [47]. Probably, the inner DysF domain is also participating in the recruitment of dysferlin to the PM [48] and an arginine-rich motif next to the transmembrane helix plays a role in PS recruitment to the sarcolemma lesions [49]. Thus, most of the studied ferlins domains have demonstrated an ability to interact with negatively charged membrane phospholipids and binding can be enhanced by Ca^2+^. Predominating of such domains in the ferlins structure, positions of the proteins with regard to membranes (their topology) and localization of the target negatively charged lipids in the inner leaflet of PM have likely important consequences for their function. The topology of a tail-anchored protein with a large cytoplasmic domain, a single transmembrane domain at the C-terminus and a small luminal or extracellular domain, together with Ca^2+^-sensitivity make it likely that all ferlins directly operate in Ca^2+^-regulated intracellular membrane fusion and trafficking. As the functional part of the ferlins is oriented towards the cytoplasm, this may mean that cell-to-cell fusion and syncytia formation are only indirectly affected [50]. Of note, one of the ferlins partner negatively charged lipids, namely PS, is actively transported to the outer leaflet of the PM in cells, including myoblasts, before syncytia formation [51]. However, there is no evidence that ferlins can invert their topology to orient the MC2D part to the extracellular compartment.

Unfortunately, 3D structures of full-length ferlins are still unknown, as it is difficult to isolate the full-length proteins. To date, structures of the C2A domains of otoferlin, dysferlin and myoferlin [42,52,53], of FerA domains of the same proteins [53], and of the inner DysF domains of human dysferlin and myoferlin have been resolved [54,55]. All known domain structures represent approximately 15% of the dysferlin protein (C2A 101 amino acids (aa), FerA 112 aa and inner DysF 109 aa).

## 5. Tissue Distribution of the Ferlins

Dysferlin is ubiquitously expressed in human tissues [33,56]. Myoferlin is produced in muscle, heart, lung [57], airway epithelia [58], vascular endothelia [59], placenta [60], skin, testis and in several cancer tissues [61,62,63]. Human otoferlin mRNA was not detected in skeletal muscle and kidney among twelve tissues tested [33], but the corresponding protein was found predominantly in the genuine sensory cells of the mammalian cochlea, the inner hair cells (IHCs), as well as in the vestibular hair cells and in the brain [64]. The outer hair cells (OHCs) express otoferlin during a short developmental phase only [25]. In zebrafish tissues otoferlins a and b transcripts were detected in the sensory hair cells of the inner ear, otoferlin a in the mid-brain and retinal ganglion cell layer, whereas otoferlin b was in the hair cells of the lateral line [26]. Initially, *Fer1L4* long noncoding RNA was found to be selectively transcribed in human stomach tissue [33], however, later Fer1L4 was detected in multiple normal tissues surrounding malignant tumors [65,66,67,68]. Fer1L5 is produced in myotubes [69], pancreas and at lower levels in a few other human tissues [33]. Fer1L6 transcripts are predominantly found in human kidneys, stomach and heart, however, were not detected in the human skeletal muscle [33], but was found in the mouse C2C12 myoblast line, in gills and gonads of adult zebrafish and broadly distributed in the head and trunk of zebrafish larvae [34].

Although dysferlin mRNA and protein demonstrate a very broad tissue distribution, its mutations or absence cause rather specific disease phenotypes, affecting predominantly the skeletal muscle, which can be rescued by muscle-specific transgenic dysferlin expression in mice [70]. Similarly, otoferlin mRNA is expressed in various human tissues [33]. Nevertheless, the loss of otoferlin function in humans has a distinct disease phenotype and causes nonsyndromic sensorineuronal deafness DFNB9 affecting SV exocytosis by the cochlear IHCs [6,25].

Four of the mammalian ferlins are expressed in muscle. Three of those, dysferlin, myoferlin and Fer1L5, are type I ferlins [9]. Fer1L6 is a type II ferlin expressed in C2C12 mouse myoblasts before and after differentiation, and playing a role in zebrafish skeletal muscle development [34]. In contrast to Fer1L6, myoferlin, Fer1L5 and dysferlin are predominantly expressed at different stages of myogenic differentiation in vitro and in vivo and could be classified as mostly associated with earlier stages of differentiation (myoferlin), later stages (dysferlin) and intermediate stages (Fer1L5). Thus, myoferlin is expressed in in vitro cultured C2C12 myoblasts and its expression decreases when myoblasts differentiate into myotubes [71]. However, myoferlin expression increases in damaged myofibers of the *mdx* mouse model of Duchenne muscular dystrophy (DMD) and in DMD patients biopsies [57,72,73]. Myoferlin mRNA is also upregulated in mature muscles upon resistance exercise training [74]. Fer1L5 is expressed in myotubes containing 2–4 nuclei, and its level decreases upon further growth of the myotubes [69]. Interestingly, dysferlin expression increases during myogenic differentiation in vivo and in vitro and persists in the mature skeletal muscle. Dysferlin can be detected already in activated MyoD-positive satellite cells in human skeletal muscle biopsies, and its level increases in multinucleated myotubes [75]. Prominent elevation of dysferlin levels was observed in in vitro differentiating C2C12 cells [76] and its expression continues in mature myofibers in vivo [56]. Thus, in development, dysferlin and myoferlin have the opposite expression dynamics, however, similarly to myoferlin, dysferlin expression demonstrates a moderate increase (approximately 2-fold) in biopsies of DMD patients and a four-fold increase in the corresponding mdx mouse model in comparison to age-matched wild-type controls [77].

Distribution of dysferlin among the PM compartments (transverse (T)-tubules and sarcolemma) is changing during muscle maturation. In immunohistochemistry experiments, the intensity of dysferlin staining peaks on T-tubules during their development and regeneration, but redistributes predominantly to the sarcolemma in the mature muscle fibers [56,77,78], without losing the staining of T-tubules [79]. Dysferlin association with T-tubules and sarcolemma was confirmed by subcellular fractionation and microscopy [80,81,82]. Accordingly, these alterations in the ferlins repertoire and localization can have important consequences for the skeletal muscle integrity and differentiation.

In the following three sections (Section 6, Section 7 and Section 8), we discuss the main functional aspects of the muscle ferlins dysferlin, myoferlin, Fer1L5 and Fer1L6 in turn.

## 6. Functions of Dysferlin in Muscle

### 6.1. Dysferlin Functions in Sarcolemma Repair

The best-studied function of dysferlin is its role in repair of lesions in the surface membrane of striated muscle fibers, the sarcolemma [83]. Muscle fiber contraction mechanically stresses the sarcolemma resulting in micro-lesions. These need to be repaired quickly and efficiently to prevent leakage and death of damaged muscle fibers. The repair process is triggered by Ca^2+^-influx into the sarcoplasm through the lesion and depends on a set of proteins including dysferlin as one of the key players [83,84]. It is likely, that dysferlin exerts its role during membrane repair promoting membrane aggregation and fusion via its Ca^2+^-triggered interactions with negatively charged phospholipids [43,45,83]. Dysferlin trafficking and dysferlin-dependent membrane repair are supported by partnering proteins. These include:Ca^2+^- and PS-binding proteins annexins A1, A2 and A6 [85,86];Muscle-specific proteins mitsugumin 53 (MG53) and caveolin 3, which are important for the nucleation of the sarcolemma repair machinery and for regulating the trafficking of dysferlin to and from the PM, respectively [87,88];A giant scaffolding protein AHNAK participating in the regulation of Ca^2+^ homeostasis, signaling and structure of cytoskeleton [89,90];Myoferlin, another member of the ferlin protein family [91];Affixin (β-parvin), a protein linking integrins and cytoskeleton [92]; and;A focal adhesion protein vinculin, cytoplasmic dynein participating in the retrograde vesicle transport along the microtubules and tubulin A [91,93].

These dysferlin-interacting proteins link its function as a Ca^2+^-sensitive membrane-binding protein important for vesicular trafficking during sarcolemma repair to cytoskeleton remodeling. It is likely that dysferlin is participating not only in exocytosis of vesicles or organelles dedicated for sarcolemma repair [94], but also in concomitant endocytosis [2,82]. More than that, in vivo function of dysferlin in sarcolemma repair extends to PS sorting to the site of membrane damage leading to the recruitment of macrophages, which removes the patch or plug (see below), as it was shown in zebrafish [49].

A number of cellular mechanisms have been shown to contribute to PM repair: contraction of membrane wounds, plugging (protein-based crosslinking of intracellular vesicles or membranous organelles without their fusion), patching (restoration of PM integrity by fusion of intracellular vesicles), endocytosis and externalization or membrane shedding [84,95]. The mechanisms could coexist and participate in resealing of the same PM lesion depending on the cell type or the stage of myogenic differentiation and on the extent of the PM injury. Most if not all of these mechanisms could contribute to sarcolemma resealing and proceed to a certain degree dependently on dysferlin. 

Relatively large (up to 4 µm) sarcolemma lesions of mature myofibers could be resealed by one of the two mechanisms called patching and plugging, while fusion of the membranous organelles within the patch or plug was not proven. Formation of a dysferlin-containing patch or plug on the sites of sarcolemma wounds in zebrafish was paralleled by an increase in a PS-sensor signal and BODIPY-cholesterol fluorescence, confirming the presence of membranous organelles in the patch or plug [49,96]. These data are supported by earlier observations of vesicle accumulation below membrane lesions in non-necrotic muscle fibers from biopsies of dysferlinopathy patients and dysferlin knock-out mice [83,97,98]. These findings point to a defect in vesicle aggregation or fusion in the absence of dysferlin or under the conditions of severe reduction of its level. Resealing of smaller lesions (≥120 nm) also requires dysferlin, but without the formation of a dysferlin-containing patch [86]. It is likely that in such cases the repair of sarcolemma wounds requires formation of a proteinaceous plug or repair cap made up of several annexins. Indeed, when fluorescently labeled, dysferlin was not found in the repair cap, which was also devoid of the negatively charged lipids PS and PIP2, questioning the presence of the membranous organelles in the cap. During the repair process, dysferlin accumulates around the repair cap in a ‘shoulder’ area, possibly via lateral diffusion within the sarcolemma [49,82,86] and its interactions with the cytoskeleton, recruiting PS and hence macrophages to the injury sites [49]. The fusion of dysferlin-containing vesicles with the shoulder regions in this repair process was not demonstrated yet, but could not be excluded.

Nevertheless, the function of dysferlin in vesicular trafficking, which could underlie its role in PM repair, is supported by the observed defects in the injury triggered lysosome exocytosis across the surface of dysferlin-deficient myotubes and myoblasts [85,94]. Of note, dysferlin does not localize to lysosomes in intact myotubes, but dysferlin-containing vesicles fuse with lysosomes upon sarcolemma damage [99]. The lysosomal exocytosis could serve at least two functions: (i) acid sphingomyelinase secretion, which promotes membrane invagination and endocytosis, e.g. of caveolae [100,101,102] and (ii) it is likely that rather uniform secretion of lysosomal enzymes along the surface of the damaged muscle fiber could digest the basal lamina surrounding it and thereby reduce the mechanical stress on the fiber. The origin of the organelles that form the repair patch or fuse next to the site of sarcolemma injury has not been identified yet. Candidate compartments are T-tubule derived vesicles and vesicles originating from the sarcolemma or its subcompartments caveolae as well as enlargosomes [101,103,104]. 

Another mechanism of sarcolemma repair, which could depend on the function of dysferlin, is the contraction of the membrane wounds. This idea is supported by several observations: (i) dysferlin is accumulating on the rims of the lesions likely by lateral diffusion [49,82,86,105], (ii) the process of lesion contraction is Ca^2+^-dependent [105], (iii) dysferlin C2-domains are Ca^2+^-sensitive [43], (iv) dysferlin directly or indirectly interacts with cytoskeleton [89,91,92,93,106] and (vi) dysferlin recruitment to a wound site is dependent on annexin A6, a protein that previously has been shown to be involved in membrane lesion constriction in another cell type [86,107].

In summary, dysferlin bears a potential to participate in sarcolemma repair by at least four mechanisms (Figure 4):
Membranous repair patch or plug formation;T-tubule stabilization (see below) with T-tubule as a possible membrane reservoir;PS-sorting; recruitment of macrophages and contraction of the membrane wound, and;Lysosome exocytosis.


### 6.2. Dysferlin Functions in Triad Biology

Besides its function in the repair of sarcolemma of striated muscle fibers, dysferlin plays a role in T-tubule system development and in triad function upon injury. It participates in sculpturing the membranes during T-tubule biogenesis, especially in regenerating muscles and possibly also during repair of the system upon injury [1,77]. More than that, dysferlin takes part in the regulation of Ca^2+^-metabolism of injured muscle fibers via mechanochemical stabilization of the triad junction and its Ca^2+^-release and thus decreasing triad and, in particular, T-tubule sensitivity to stress [78,81,108,109].

Anatomically, the triad is defined by three membrane compartments: one T-tubule in the center, surrounded by two terminal cisternae of the sarcoplasmic reticulum. The main function of the triad is excitation–contraction coupling of the striated muscle fibers, which is achieved by physical binding of voltage-gated Ca^2+^ channels (Ca_v_1.1), also known as L-type Ca^2+^-channels (LTCC) or dihydropyridine receptors (DHPRs) localized to T-tubules, and the ryanodine receptors (RyRs), calcium channels mediating calcium-induced calcium release from terminal cisternae of sarcoplasmic reticulum.

Dysferlin was found in a complex with DHPRs and caveolin 3, and also with RyRs [80,93]. Dysferlin possibly stabilizes the Ca^2+^-metabolism of the injured muscle fibers inhibiting DHPRs and preventing injury-induced Ca^2+^-leak into the sarcoplasm through RyRs, model supported by sustained Ca^2+^ influxes in dysferlin-deficient muscle fibers sensitive to the DHPR inhibitor diltiazem and reduction of the extracellular Ca^2+^ concentration [81]. However, later the sarcoplasmic reticulum and RyRs were identified as the primary source of the Ca^2+^ leak in injured muscle fibers in the absence of dysferlin [109]. Thus, dysferlin localized to T-tubules forms a complex with both DHPRs and RyRs and modulates their function in the case of injury.

Dysferlin-deficient muscles also show T-tubule system abnormalities upon regeneration [1]. Dysferlin interacts with the T-tubule proteins caveolin 3 and amphiphysin 2 as well as with negatively charged lipid PIP2 required for T-tubule biogenesis [1,45,110,111]. It was shown that dysferlin induces the formation of tubular structures upon expression in non-muscle cells [1]. Otoferlin and myoferlin did not induce such intracellular membranes when overexpressed under similar conditions. When truncation mutations and pathogenetic point mutations in dysferlin were analyzed, all except one (L1341P located in C2E) failed to induce membrane tubulation in non-muscle cells. In C2C12 myoblasts, dysferlin colocalizes with PIP2 at the PM and at the T-tubule system. Dysferlin also binds to PIP2-rich vacuoles that were generated by expression of phosphatidylinositol phosphate kinase or a constitutively active Arf6 mutant [1]. Interestingly, most analyzed dysferlin deletions and truncations were not recruited to PIP2-vacuoles, indicating that PIP2 binding is a critical feature of dysferlin. Moreover, when cellular PIP2 have been degraded, dysferlin was lost from the T-tubule system, the T-tubule system was altered and, in non-muscle cells, dysferlin overexpression, could no longer induce formation of T-tubule-like intracellular membrane structures [1]. These data suggest that dysferlin together with PIP2 is critical for the biogenesis of the T-tubule system. In addition, the repair of the T-tubule system could be mediated by sarcolemma resealing complex of dysferlin, MG53 and annexin A1, as these proteins are enriched at longitudinal tubules of the system under overstretch conditions [77].

### 6.3. Dysferlin in the Differentiation, Growth and Regeneration of Skeletal Muscle 

Skeletal muscles develop normally in pre-symptomatic dysferlinopathy patients and dysferlin-deficient mice. However, after reaching a certain age, several muscle groups begin to show initial signs of pathology: centrally nucleated fibers and variability in fiber size, which deteriorate with time. So, if dysferlin plays a role in muscle growth, myoblast differentiation and fusion, then it, likely, takes place after a defined developmental stage. There are contradicting reports concerning dysferlin role in skeletal muscle differentiation, grow and regeneration. Some authors present evidence for the absence of an effect of dysferlin-deficiency on myoblast differentiation in vitro [94,112,113]. However, none of them presented fusion indices or the numbers of nuclei per myotube differentiated in culture. There are also reports about higher regeneration capacity upon injury in mice bearing a mutation in *Dysf* [114,115,116]. Other authors argue for a role of dysferlin in myoblast differentiation and cytokine secretion so that myoblast fusion is affected indirectly only. In their studies, dysferlin-deficiency leads to decreased levels of myogenesis regulatory factors like MyoD and myogenin and delays myogenic differentiation in vitro [117,118]. Accordingly, induction of dysferlin expression in myoblasts was shown to promote their myogenic differentiation [76]. Myoblasts isolated from dysferlinopathy patients or derived from dysferlin-deficient mice proliferated with the normal rate [117,118], but showed decreased fusion efficiency in vitro as a result of activated signaling of the pro-inflammatory network inhibiting myogenesis [118]. In this context it is important to mention that dysferlin has been found in a protein complex with minion/myomerger, a fusogenic protein, which together with myomaker conveys the ability to form syncytia to myogenic and non-myogenic cells [119].

Impaired adult satellite cell differentiation, myoblast-to-myotube fusion and muscle growth in the absence of dysferlin can be attributed to a defect in insulin-like growth factor-1 receptor (IGF1R) trafficking, since IGFs are known to promote muscle cell differentiation [120,121]. Furthermore, dysferlin-deficiency attenuated muscle regeneration resulting in the presence of an increased number of immature fibers and suggesting that regenerative process is delayed or incomplete in dysferlinopathy [122]. Normal muscle regeneration process requires temporally acute and transient immune response for well-timed removal of necrotic fibers [123], however, the extended inflammatory response in a mouse model of dysferlinopathy is probably due to the defect in stimulated cytokine release by myoblasts [122].

In summary, dysferlin could have additional functions in vesicular trafficking of growth factors receptors, secreted pro-inflammatory molecules and even fusogenic proteins that promote muscle growth and regeneration. We hypothesize that dysferlin-dependent trafficking of such signaling molecules can modulate gene expression and the function of the adult muscle stem (or satellite) cells responsible for the skeletal muscle growth and regeneration in mature individuals. 

## 7. Functions of Myoferlin and Fer1L5

Like dysferlin, myoferlin and Fer1L5 are important for skeletal muscle growth [124,125,126]. Myoferlin can localize to the PM and intracellular vesicles in myoblasts and to the sarcolemma of mature muscle fibers [33,57,124]. However, in contrast to dysferlin, myoferlin was not found associated with T-tubules and does not induce tubular structures in cells upon heterologous expression [1,108]. Myoferlin also can localize to the nuclear envelope and translocates to the nucleus together with the transcription factor STAT3 upon activation [57,127]. Other myoferlin interacting proteins include dysferlin, AHNAK, ADAM12 (A Disintegrin and Metalloproteinase 12) and EHD1 and 2 (Eps15 homology-domain containing proteins 1 and 2 regulating endocytic recycling) [69,89,91,128].

The first function described for myoferlin was its role in skeletal muscle growth and regeneration [124]. Interestingly, myoferlin knock-out mice have lower body mass with decreased diameters of skeletal muscle fibers. These mice show delayed muscle regeneration upon cardiotoxin injection, but no myopathy [124]. In contrast, dysferlin-deficient mice grow normally up to a certain age and later develop muscular dystrophy [69], emphasizing functional differences between myo- and dysferlin in mice. In humans for a long time no pathogenic mutations in *MYOF* was reported. However, recently the first case of limb-girdle type muscular dystrophy and associated cardiomyopathy linked to *MYOF* mutation was described [7]. 

Known Fer1L5-binding proteins are EHD1, EHD2 and GRAF1 (Rho-GAP GTPase regulator associated with focal adhesion kinase-1) [69,129,130]. EHD proteins are dynamin-related ATPases capable of vesicle scission, while GRAF1 regulates the actin cytoskeleton and sculptures membranes [130,131]. It was suggested that myoferlin and Fer1L5 mediate intracellular trafficking events essential for efficient myoblast to myotube fusion, and that knock-down of EHD2 or GRAF1 interferes with trafficking of these ferlins to a cell periphery [69,129,130]. Indeed, together with EHD proteins myoferlin and Fer1L5 could participate in recycling of IGF1R and the glucose transporter GLUT4, which both are required for muscle growth [125,126,128]. It would be important to determine whether myoferlin, like dysferlin, could play a role in trafficking of the myoblast fusion proteins minion/myomerger and myomaker.

Myoferlin is also involved in membrane repair in muscle fibers and during accelerated proliferation of tumor cells [86,132] as well as in the maintenance of T-tubules stability and function in striated muscle [108]. Thus, myoferlin, similar to dysferlin, may be required for multiple trafficking events in the secretory and endocytic pathways and the functions of the muscle-expressed ferlins, dysferlin, myoferlin and Fer1L5, could overlap to a significant degree.

## 8. Function of Fer1L6

The available data on *FER1L6/fer1l6* point to two functions of the gene. In humans, *FER1L6* was identified as a gene linked to prostate cancer progression [133]. In zebrafish, it was found to be important for skeletal and for cardiac muscle development [34]. In addition to the muscle phenotype, fer1l6 deficiency led to general abnormalities of zebrafish larvae development (e.g., of head and eyes) and to high mortality [34]. The absence of fer1l6 in zebrafish resulted in compensatory overexpression of dysferlin and myoferlin, and significant changes in the expression of the satellite cells marker pax7 as well as markers of myogenic differentiation such as myod1 and mrf4 [34]. It is tempting to speculate that increased expression of myod1 and mrf4 and simultaneous decrease of the pax7 mRNA level indicate an imbalance between maintenance and differentiation of the satellite cells with decreased satellite cell population and an increase in the number of differentiating myogenic progenitor cells and myoblasts. This suggests at least an indirect role of fer1l6 in differentiation of satellite cells and muscle growth, e.g., via maintaining normal level of dysferlin expression [76,117,118].

## 9. Functions of Muscle Ferlins in Non-Muscle Cells

Dysferlin or myoferlin knock-out in mice leads to late onset muscle dystrophy or a defect in skeletal muscle growth, respectively [83,124]. However, as these ferlins have broad tissue distribution, their functions in cell types other than skeletal muscle may include:Mediating lysosome exocytosis in endothelial cells (dysferlin) and phagocytes, or growth factors (vascular endothelial growth factor A, VEGFA) exocytosis in pancreas cancer cells (myoferlin) [134,135,136];Recycling of endocytosed transferrin and IGF1Rs in fibroblasts (dysferlin), vascular endothelial growth factor receptor 2 (VEGFR 2) in endothelial cells and signaling of epidermal growth factor receptor (EGFR) in breast cancer cells (myoferlin) [59,61,120];Clathrin- and caveolae-dependent endocytosis in endothelial cells (myoferlin) [137];Membrane repair in tumor and immortalized cells (myoferlin) and, potentially, in the placenta (dysferlin), [132,137,138];Regulation of activity and contact formation of such cells of the immune system, as monocytes, and of adhesion in endothelial cells (dysferlin) [139,140,141];Possibly, trophoblast fusion in the placenta (dysferlin) [60].

Proteins found to interact with these ferlins in non-muscular cells are dynamin 2 (myoferlin) [59], caveolin 1 (myoferlin) [137], platelet endothelial cellular adhesion molecule-1 (PECAM-1) (dysferlin) [140], integrin β3, vinculin, paxillin and β-parvin (dysferlin) [141]. These binding partners confirm a role of dysferlin and myoferlin in the formation of focal adhesion sites and endocytosis, respectively. 

## 10. Functions of Otoferlin

Otoferlin is important for hearing in all studied bony vertebrates, its deficiency leads to profound nonsyndromic deafness in humans, profound hearing loss in mice accompanied by a mild vestibular deficit, and a hearing defect in zebrafish coexisting with a more prominent balance and locomotion dysfunction [6,25,26,142]. Hearing and balance rely on the function of hair cells—the genuine sensory cells of the cochlea, vestibular system and lateral line (the latter is present in aquatic vertebrates only). The hair cells form the first synapses of the pathways with neurons. The IHCs of the mammalian cochlea represent the primary place of expression of otoferlin in adult animals and their ribbon synapses outperform most if not all other characterized synapses. Thus, SV exocytosis at the first auditory synapse is unprecedented in such properties as speed, timely precision and indefatigability (for review [3,10]). The molecular makeup of this synapse is unique, e.g., since mature IHCs express Piccolo and RIM2 α and β, but no Munc13s or synaptotagmin I and II, among the MC2D proteins, important for SV exocytosis at the conventional central nervous system synapse [143,144,145,146,147,148]. Otoferlin is one of the key molecules determining the properties of SV exocytosis at the IHCs ribbon synapse. The role of otoferlin in membrane dynamics, i.e., in the SV cycle in the IHCs, is the best studied among ferlins. However, when interpreting the data attention has to be paid to the differences in terminology and the absence of direct correlation between functional SVs tethering, docking and priming described for the conventional central nervous system synapse and the morphological tethering and docking at the IHCs ribbon synapse. Otoferlin plays a role in multiple steps of the SV cycle in the IHCs: (i) functional docking, (ii) priming; (iii) fusion, (iv) endocytosis and, possibly, (v) preceding the later transport of the material dedicated for endocytosis to the sites of recycling near the active zone and, lastly, (vi) maturation of SVs.

In the otoferlin knock-out mice, the average tether length between SVs and active zone increases from 20 to 30 nm [148]. This observation can be interpreted as looser attachment of SVs to the PM in the absence of otoferlin, and as a defect in the transition from functional tethering to docking, or from docking to priming. SV priming is, likely, impaired by the otoferlin missense mutation *pachanga* (D1767G) [149]. The downstream Ca^2+^-triggered SVs exocytosis is almost completely blocked in the IHCs of otoferlin knock-out mice [25]. Following the fusion of SVs with PM, the clearance of the active zone including the efficient transport of the material away from the places of exocytosis and endocytosis per se are dependent on otoferlin and, likely, on its interaction with endocytic adaptor proteins AP-2µ and endophilin-A1, motor protein myosin VI, and GTPase Rab8b [149,150,151,152,153,154,155] (reviewed in [3]). It was shown, that SVs on average have larger diameters in a temperature-sensitive (I515T) and *pachanga* mutants, indicating a defect in the reformation/maturation of SVs and thus, a role of otoferlin in this process [156,157]. The functions of otoferlin in the IHCs could be regulated by phosphorylation, e.g., changing Ca^2+^-sensitivity of C2C and C2F domains [158]. In summary, otoferlin not only determines the mode of exocytosis [159], but also participates in multiple steps of the SV cycle in the IHCs.

## 11. Fer1L4—A Non-Muscle Ferlin Long Non-Coding RNA

In humans, the long non-coding RNA Fer1L4 plays a role in signaling, controlling cell proliferation, migration and apoptosis. The results of the vast majority of the studies are consistent with a function of Fer1L4 as a tumor suppressor in multiple types of cancer [32,66,67,68,160,161]. The exception is a report about the Fer1L4 possible role in glioma progression, where high expression of Fer1L4 was associated with a poor disease prognosis [162]. Fer1L4 exerts its function via the PTEN/PI3K/AKT signaling pathway [161,163,164]. The role of this pathway in tumorigenesis was described in detail [165]. PTEN is known as a tumor suppressor inhibiting PI3K/AKT proliferation signaling. Downregulation of Fer1L4 expression correlated with decrease of the level of PTEN [164,166], while Fer1L4 overexpression inhibited the PI3K/AKT signaling pathway [67,163]. Thus, the Fer1L4 RNA may serve as a competing endogenous RNA and regulate the expression of PTEN via miRNA-mediated mechanisms inhibiting cancer cell proliferation and metastasis [164,166].

## 12. Mechanism of Action of Ferlins in Membrane Fusion

One of the most intriguing questions concerning ferlins is their mechanism of action in Ca^2+^-triggered fusion of membranous organelles. Is the function of ferlins SNARE-dependent? Can ferlins act as Ca^2+^-sensitive core fusogens and fuse membranous organelles in the secretory and lysosomal pathways independent of other proteins? Or do ferlins function as Ca^2+^-sensors in SNARE-dependent membrane fusion, analogous to synaptotagmins? Are the C-terminal fragments comprising of two C2 domains and the transmembrane domain that result from calpain cleavage of ferlins the core functional modules? [105,167,168]. Are the full-length ferlins organizing SNAREs and Ca^2+^-channels next to a prospective fusion pore? [2].

These questions require a short introduction into Ca^2+^-triggered SNARE- and synaptotagmin-dependent membrane fusion, because in such membrane fusion reactions Ca^2+^-sensitivity is provided by two-C2-domain containing proteins, the synaptotagmins (Figure 1) and the energy for the fusion by zippering of SNARE-protein complexes. Among the best-studied SNARE-complexes are the neuronal SNARE-complex, consisting of SNAP25, syntaxin 1 and VAMP2 proteins [169], and the ubiquitously expressed SNARE complex comprising SNAP23, syntaxin 4 and VAMP2 [170]. It was suggested that otoferlin and dysferlin act as Ca^2+^-sensors similar to synaptotagmins and directly trigger SNARE-dependent membrane fusion [41,171]. Dysferlin interacts with SNAP23 and syntaxin 4 while otoferlin binds SNAP25 and syntaxin 1 in vitro [41,171,172]. However, in contrast to dysferlin, SNAP23, syntaxin 4 and VAMP4 did not accumulate at the sites of membrane lesions in vivo or in cultured myotubes [96,105]. Furthermore, none of the proteins of the neuronal SNARE-complex is involved in neurotransmitter release in the IHCs—the primary place of action of otoferlin—since neuronal SNARE proteins knock-outs or their cleavage with neurotoxins had no effect on the SV exocytosis in these cells [173]. An attempt to functionally substitute otoferlin with synaptotagmin I and vice versa in the IHCs, chromaffin cells and hippocampal neurons did not result in rescue of the knock-out phenotypes [174]. Thus, most ferlins are essential for some specialized membrane fusion events [25,69,83,125,175], but whether these are dependent on SNARE proteins is not yet clear.

The current literature discusses controversially the question if full-length ferlins are the functional executioners of ferlin action, or if ferlin function can be taken over by partial proteins so that full-length ferlins are just precursors for a functional two-C2 domain module [2,105,167,168,176]. First of all, disease-causing missense mutations are uniformly distributed along the length of the ferlin genes and proteins, supporting a role of the full-length proteins (e.g., dysferlin [177], Table 1). However, these mutations could cause unfolding and degradation of the ferlins, like in the case of *pachanga* otoferlin mutation [149] or dysferlinopathies (the latter are diagnosed on the basis of severe reduction in the corresponding protein level). However, recently a role of the full-length ferlins in the Ca^2+^-triggered membrane fusion events was supported by additional data on otoferlin [178]. The authors generated transgenic mice carrying mutations in C2C domain (D515A and D517A), decreasing its Ca^2+^-sensitivity. The level of expression of the mutated otoferlin was not changed, but the mutations had functional consequences for SV cycle and hearing. In summary, these results present evidence for the functional importance of the ferlin N-terminus, domains other than C-terminal C2F and G (Figure 1).

The importance of ferlins is associated with their involvement in human disease. Therefore, in the remaining three sections (Section 13, Section 14 and Section 15), we will discuss clinical conditions associated with genetic defects and alterations in expression levels of dysferlin, otoferlin and myoferlin.

## 13. Ferlins in Human Diseases: Dysferlinopathies and Their Pathomechanisms

Dysferlinopathies are diseases caused by mutations in *DYSF*, affecting mainly skeletal muscles [180]. There are two common dysferlinopathy phenotypes—limb girdle muscular dystrophy type 2B (LGMD2B) and Miyoshi myopathy (MM)—along with several more rare conditions: distal myopathy with anterior tibialis onset (distal anterior compartment myopathy), congenital muscular dystrophy and isolated hyperCKemia, an elevated concentration of serum creatine kinase (CK) [180]. Onset and progression of the disease as well as distribution of muscle weakness and wasting may vary significantly between individuals affected by dysferlinopathies. Several different clinical phenotypes can occur even within families carrying the same pathogenic variants of *DYSF* [5,181,182,183]. These observations emphasize the importance of investigating potential modifier genes [5,181].

Dysferlinopathies are characterized by late onset and slow progression. In carriers of pathogenic gene variants disease usually manifests in the second or third decade of life. The first symptoms are lower limb weakness accompanied by an increase in serum CK levels. The patients with the most severe phenotype of LGMD2B can become confined to wheelchair after two or three decades of disease progression, while most MM patients preserve ambulation [184]. Histological signs of the diseases are degeneration and regeneration of skeletal muscle [185] and in the more severe cases, fibrotic and adipogenic replacement of myofibers. On the protein level, dysferlinopathies are diagnosed by a complete loss or severe reduction of dysferlin in muscle biopsies or peripheral blood monocytes [186].

Mouse models lacking dysferlin develop muscular dystrophy, however, with an apparently less severe phenotype than humans and do not lose ambulation with age [187,188]. In dysferlin-deficient mice, the earliest symptoms are centrally nucleated fibers and marked differences in the myofiber diameter as well as 4- to 6-fold elevated CK levels at four weeks of age [189].

Latent cardiac dysfunction has been reported in dysferlinophathies, but patients do not primarily suffer from cardiomyopathies [190,191]. In a retrospective analysis, cardiac and respiratory functions were studied in dysferlinopathy patients [191]. Thus, a cardiovascular magnetic resonance analysis of LGMD2B patients showed indications of mild structural and functional cardiomyopathy [192]. One fifth of the patients developed respiratory problems and 9% required non-invasive ventilation. Accordingly, heart function in dysferlinopathy mouse models is either not or mildly affected [187,190,193,194,195]. However, also membrane repair in cardiomyocytes is dependent on dysferlin [196] and in a model of ischemia/reperfusion injury, dysferlin was shown to be cardioprotective [196,197]. Furthermore, physical stress exercise, or β-adrenergic activation provoke various symptoms of cardiac dysfunction in dysferlinopathy mice [190,194,196,197,198,199]. Mice with dysferlin inactivation show increased susceptibility to coxsackie virus infection and virus-induced myocardial damage [200], suggesting that pathways of viral infection and muscle repair may overlap.

It is generally assumed that disease causing mutations are more or less uniformly distributed along the dysferlin-coding sequence [177] (Table 1). Two pathogenic missense mutations in human dysferlin FerA do destabilize the domain in differential scanning calorimetry (DSC) experiments [35] and a similar prediction was made for the three most frequent out of 15 missense mutations in the inner DysF domain [55]. These and other mutations may result in the poor dysferlin folding and degradation of the protein via different pathways. For example, missense mutation L1341P in C2E domain causes dysferlin aggregation in the ER and degradation by the additional autophagy/lysosome ER-associated degradation system [201]. Dysferlin lacking C2C domain or carrying patient mutation L344P within FerI domain demonstrate accelerated endocytosis, protein lability and endosomal proteolysis [47].

At present it is not clear how exactly loss-of-function mutations of *DYSF* and a decrease of the corresponding protein expression level lead to the development of dysferlinopathies. The following mechanisms may contribute to the development of the disease: (a) a defect in sarcolemma repair; (b) changes in Ca^2+^-homeostasis; (c) impaired muscle growth and regeneration and (d) inflammatory processes. Below, we discussed these factors in turn. To which degree, however, these mechanisms contribute to the development of the disease is still not known. The situation becomes even more complex when considering the existence of the various clinical manifestations of dysferlinopathies.

### 13.1. Defective Repair of Myofiber Sarcolemma and, Possibly, T-Tubules

The pathomechanism could be the following: dysferlin deficiency decreases the efficiency of sarcolemma and, probably, T-tubule system repair [77,83,202]. This increases (i) influx of Ca^2+^ into injured muscle fibers, (ii) leakage of the muscle fiber contents such as muscle enzymes, e.g., CK, and (iii) the probability of death of damaged myofibers [5,83]. The latter promotes cycles of muscle degeneration and regeneration. In parallel, inefficient sarcolemma repair changes the properties of the regenerative niche by enhanced accumulation of dysferlin partner protein annexin A2 in the myofiber matrix [86,203]. The formation of a regenerative niche requires the absence of annexin A2 in the myofiber matrix, and if it is progressively accumulating, fibro/adipogenic precursors are escaping apoptotic signal. This leads to their differentiation into adipocytes and the substitution of muscle fibers in dysferlinopathy [203].

### 13.2. Changes in Muscle Fibers Ca^2+^ Homeostasis

Overloading of the cells with Ca^2+^ or abnormal intracellular distribution of these ions can lead to autophagic, necrotic or apoptotic cell death [204]. In muscles of dysferlinopathy patients, altered Ca^2+^ homeostasis and Ca^2+^-mediated cytotoxicity can result from (i) impaired sarcolemma and T-tubule system repair contributing to the leakage of extracellular Ca^2+^ into the sarcoplasm through a lesion and DHPRs as well as sarcoplasmic reticulum-stored Ca^2+^ through RyRs, (ii) abnormalities in the biogenesis of the T-tubule system and triads as well as a decrease in their plasticity in response to stress and (iii) enhanced X-ROS (NADPH oxidase 2-dependent reactive oxygen species, ROS) signaling activating mechano-sensitive Ca^2+^ channels in the T-tubule system, coupling mechanical stress to changes in the intracellular Ca^2+^ concentration [205].

Under normal physiological conditions, ROS production is linked to both signaling and metabolism (as a side product of the latter) [206]. In the mouse model of dysferlinopathy (A/J strain), X-ROS signaling is amplified and contributes to the development of myopathy in aged animals (> 6 month). Thus, in the stretched dysferlin-deficient muscle fibers of A/J mice, intracellular ROS and Ca^2+^ concentrations increased in comparison to wild-type controls [205], implying that X-ROS signaling could be enhanced as a result of dysferlin dysfunction.

Altered Ca^2+^ homeostasis in dysferlinopathy can lead to myofibers death and cycles of regeneration via (i) activation of endonucleases, phospholipases and proteases like calpains leading to unwanted cleavage of cellular components; (ii) triggering multiple signaling cascades affecting gene expression or cell survival [204].

### 13.3. Impaired Muscle Growth and Regeneration

The potential effect of dysferlin deficiency on muscle growth and regeneration was discussed above in Section 6.3. Additionally, dysferlin functions in these processes could be linked to T-tubule development in regenerating muscle, abnormalities in Ca^2+^-homeostasis and inflammatory processes (Figure 5). Formation of an irregular T-tubule network upon myofiber regeneration may disturb Ca^2+^-homeostasis. The impaired Ca^2+^-compartmentalization and signaling may lead to myofiber death, promote cycles of muscle regeneration and reduce secretion of cytokines by surviving myoblasts or myofibers [122,204,207]. The defects in secretion of chemotactic molecules leads to a decrease in the number of recruited neutrophils, delayed removal of necrotic fibers, prolonged inflammatory responses, incomplete regeneration cycles and development of muscular dystrophy [122].

### 13.4. Inflammatory Processes

Dysferlinopathies are often accompanied by muscle inflammation and dysferlinopathy patients can be misdiagnosed as having polymyositis [139,208,209]. The role of dysferlin in the inflammatory response was reviewed by several authors [138,210]. Inflammation could originate from:Leakage of damage-associated molecules such as annexin A2 from dysferlin-deficient myofibers [211] through sarcolemma lesions [210],Intrinsic pro-inflammatory signaling of dysferlin-deficient muscle fibers [118,212],Deregulation of cytokine secretion [122], and/or,Activation of dysferlin-deficient monocytes or macrophages [139].

However, there is also evidence that inflammation in dysferlinopathies originates autonomously within the skeletal muscle and not due to dysferlin function in other cell types. For example, it was shown by means of bone marrow transplantation that inflammation in SJL/J mice does not depend on the genotype of the leukocytes [213]. Along the same lines, transgenic mice generated from the A/J mouse model of dysferlinopathy, expressing dysferlin under a skeletal muscle-specific promoter are indistinguishable from dysferlin-sufficient mice [70]. Lastly, it was shown that macrophage infiltration is a consequence of myofiber damage and not vice versa [214].

In summary, reduced efficiency of sarcolemma and, likely, T-tubule system repair can lead to changes in Ca^2+^ homeostasis, myofiber necrosis, inflammation and cycles of regeneration followed by fibro-adipogenic substitution of the muscles, resulting in their weakness (Figure 5) [215]. However, pathomechanisms leading to the development of dysferlinopathies are, likely, not restricted to the defects in sarcolemma repair, since rescue of PM repair malfunctioning by myoferlin overexpression does not improve muscle histology [216]. This means that dysferlin functions other than sarcolemma repair are also indispensable for skeletal muscle health.

## 14. Ferlins in Human Diseases: Otoferlin and Deafness DFNB9

Mutations in *OTOF* cause one of the most common autosomal recessive nonsyndromic hearing losses, DFNB9, which can become apparent in three phenotypes. Unfortunately, at the moment there is no unified nomenclature for the manifestations of the disease. All DFNB9 phenotypes are auditory synaptopathies due to an Otoferlin deficit (ASO) or diseases resulting from dysfunction of the first auditory synapse between sensory IHCs and spiral ganglion neurons, which was evident from the analysis of DFNB9 patients and *Otof* knock-out or knock-in mice [25,156,217] ([10] for review). We suggest to classify the phenotypes as (i) the predominating severe-to-profound ASO (SPASO), resulting in prelingual hearing loss (HL), (ii) the more rare mild-to-moderate ASO (MASO), characterized by less severe hearing impairment determined by pure tone audiometry and disproportionally strong deficits in speech perception (formerly classified as auditory neuropathy (AN)), and (iii) the very rare temperature-sensitive ASO (TSASO), with symptoms similar to those of MASO, if any, when the patients are afebrile and exacerbated upon elevation in body temperature to severe or profound HL (formerly known as, amongst others, temperature-dependent AN) [6,218,219,220,221,222,223,224]. In some DFNB9 patients the defect in the IHC SVs cycle/exocytosis can be accompanied by age-progressive, likely, secondary dysfunction of the OHCs and, possibly, cochlea deficit, which may develop already after the first or second year of life [218]. In other patients, OHCs function may remain normal [219,225], reviewed in [10].

Currently, more than 160 pathogenic *OTOF* mutations underlying DFNB9 are known [224]. Concerning the genotype–phenotype correlation one can note that TSASO, the mildest condition, develops only in patients with missense mutations or in frame deletion of a single amino acid [156]. Patients with two nontruncating mutations may have MASO, while patients with the most severe phenotypes or SPASO often have premature stop codons [222].

At the moment otoferlin is considered to be the one and only Ca^2+^-sensor for SV exocytosis in mature IHCs, since stimulus–secretion coupling is eliminated in otoferlin knock-out mice [25,159,178], explaining such DFNB9 manifestation as SPASO. The impaired speech perception in MASO and TSASO involves enhanced adaptation to continuous or repetitive sound stimulations, which can be explained by impaired replenishment of the readily releasable pool of SVs [156]. Of note, recapitulation of temperature-sensitive symptoms of TSASO was not possible in mice [156], likely due to the higher adaptation of murine proteins to the changes in the core body temperature, since the switch from 36.5 to 38 °C is normal for the animals in contrast to humans [226] (Species-specific information at the Johns Hopkins University page: http://web.jhu.edu/animalcare/procedures/mouse.html).

## 15. Ferlins in Human Diseases: Cancer

Oncogenesis is often accompanied by enhanced mitotic signaling and a loss of factors controlling cell division, increased cell motility and invasiveness. Importantly, altered expression patterns of ferlin family members have been involved in cancer. Overexpression or downregulation of any human ferlin on the mRNA level and overexpression of myoferlin on the protein level may correlate with development of cancer (e.g., [32,61,227], reviewed in [8]). There appears to be a tight link between cancerogenesis and myoferlin and dysferlin function in muscle growth (proliferation and differentiation of satellite cells, respectively). In particular myoferlin is a part of a network promoting muscle growth, since (1) myoferlin knock-out in mice leads to smaller skeletal muscles due to reduced myofiber size [124]; (2) myoferlin enhances promitotic signaling by supporting recycling to the PM of the potent muscle growth stimulator IGF1R and preventing its degradation [125] and (3) myoferlin could possibly control the activity of the tumor suppressor and nuclear protein AHNAK and of ADAM12. Myoferlin and AHNAK interact and both can translocate to the nuclei [57,89,90,127,228,229]. ADAM12, another myoferlin binding protein normally expressed by myoblasts is required for myoblast fusion and contains a metalloproteinase domain that cleaves type IV collagens [230,231,232]. The latter is one of the main components of the basal lamina surrounding muscle fibers [233]. Under pathological conditions, ADAM12 can promote epithelial-to-mesenchymal transition and cancer invasiveness [232].

As a consequence, cells of several cancer types overexpress myoferlin and if this overexpression is knocked-down, decrease invasiveness or slow down proliferation [61,132,227,234]. It is tempting to speculate that the switch in expression of ferlins during myoblast differentiation from myoferlin in undifferentiated myoblasts to dysferlin in myotubes and myofibers might reflect the required switch from promoting proliferation by myoferlin to the promotion of myotube differentiation by dysferlin. Indeed, downregulation or a loss of dysferlin expression can be linked to impaired satellite cell differentiation and the development of rhabdomyosarcoma [117,118,235]. Thus, a link is emerging between multiple roles of dysferlin and myoferlin in conventional growth and regeneration of skeletal muscle and in oncogenesis, confirming the reprogramming of normal developmental processes in cancer.

## 16. Conclusions

As members of the superfamily of MC2D proteins, ferlins mediate both exo- and endocytosis of vesicles or organelles. Thereby ferlins play important roles in several aspects of human health, including locomotion and hearing. Diseases resulting from ferlin dysfunction are dysferlinopathies, mainly the muscular dystrophy LGMD2B and Miyoshi myopathy, nonsyndromic recessive deafness DFNB9 and also several types of cancer. Pathomechanistic models of dysferlinopathies include the interplay between defects in sarcolemma repair, Ca^2+^ homeostasis, muscle growth and regeneration and inflammatory processes. DFNB9 phenotypes originate from defects in SV cycle/exocytosis at the first auditory synapse, while cancerogenesis can reprogram and employ normal functions of ferlins in cell proliferation and differentiation.

## Figures and Tables

**Figure 1 cells-09-00534-f001:**
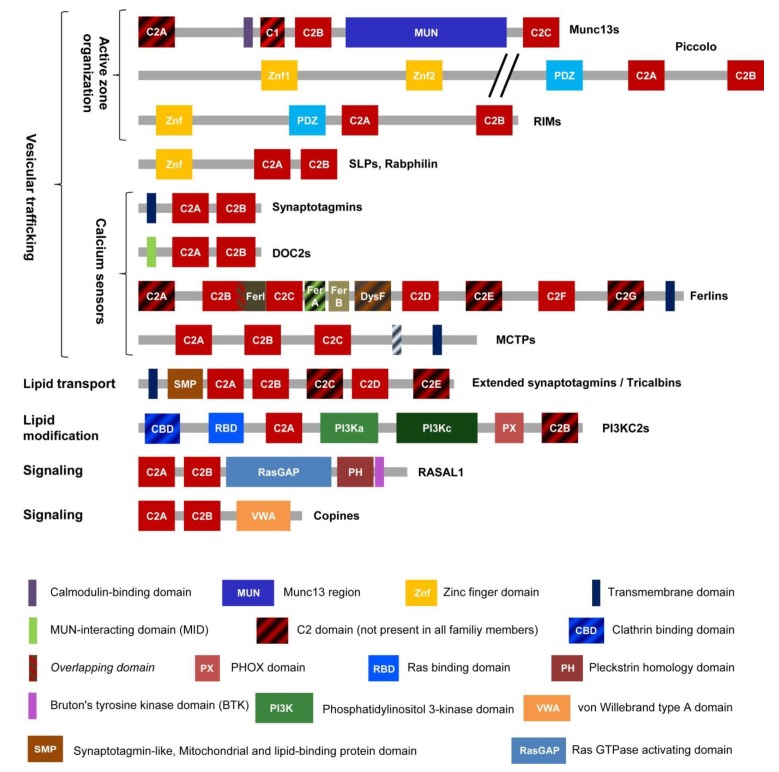
Domain organization of MC2Ds proteins. MC2Ds protein superfamily includes at least twelve protein families: Munc13s (mammalian uncoordinated-13), Piccolo, RIM (Rab3-interacting molecule), SLPs (synaptotagmin-like proteins), DOC2s (double C2 domain proteins), ferlins, MCTPs (multiple C2 domain proteins with two transmembrane regions), extended synaptotagmins, PI3KC2s (phosphoinositide 3-kinases class II; a, accessory, c, catalytic domain), RASAL (Ras GTPase- activating-like protein) and copines. The striped domain pattern designates domains present not in all family members.

**Figure 2 cells-09-00534-f002:**
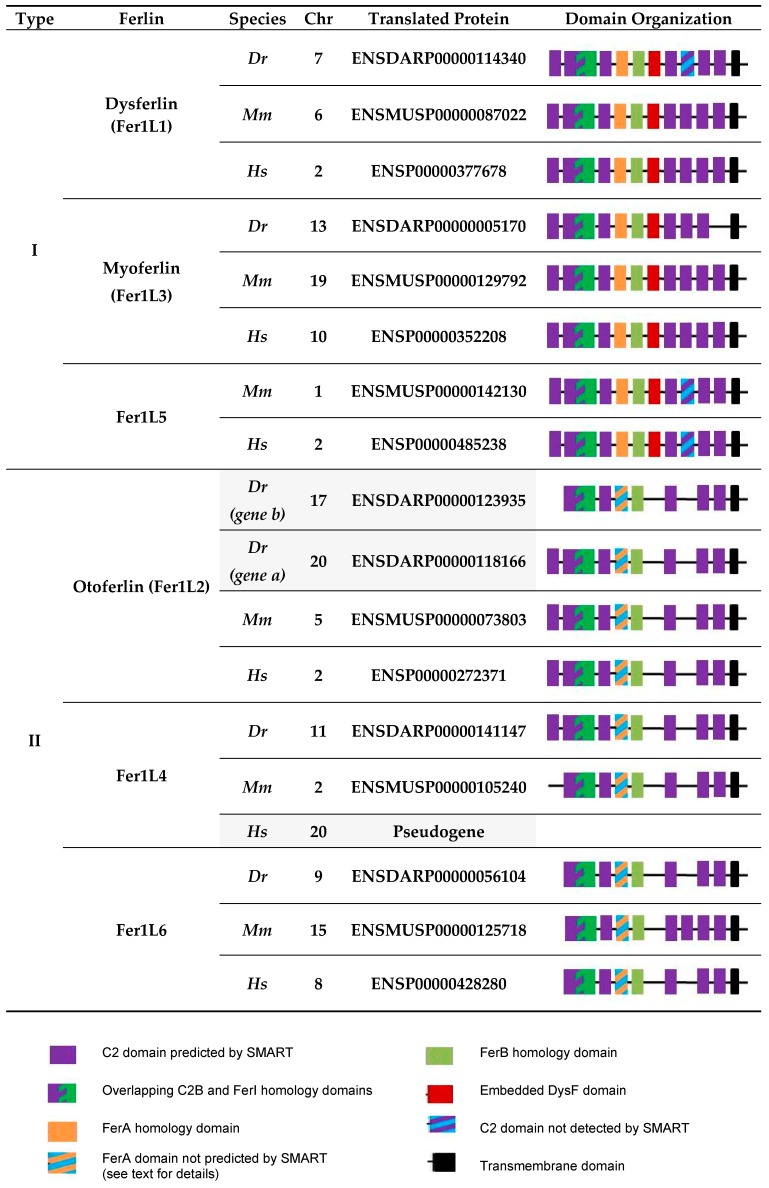
Domain organization of ferlins of bony vertebrates. Ferlin domain organization from selected species (Dr, *Danio rerio; Mm, Mus musculus; Hs; Homo sapiens)* using the genome browser Ensembl (Release 96 from April 2019) [36] was drawn according to SMART and Pfam [37,38]. The corresponding phylogenetic tree was produced using Clustal Omega multiple sequence alignment program using default parameters [39]. Translated proteins are from e!Ensembl. Zebrafish has 6 ferlin genes; *fer1l5* is absent; however, two related otoferlin genes *otofa* and *b* are present. In the mouse, all 6 ferlin genes are present and encode proteins, whereas in humans, *FER1L4* represents a pseudogene producing long noncoding RNA. Abbreviations: Chr, chromosome.

**Figure 3 cells-09-00534-f003:**
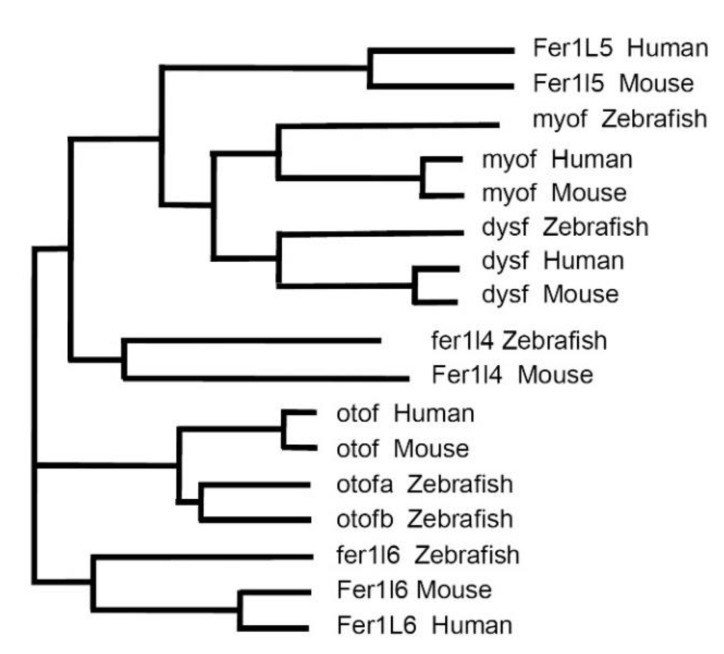
Phylogeny of ferlins in humans, mouse and zebrafish. The phylogenetic tree was produced using Clustal Omega multiple sequence alignment program using default parameters [39] and translated protein sequences from Figure 2. The branch length is indicative of the evolutionary distance between the sequences.

**Figure 4 cells-09-00534-f004:**
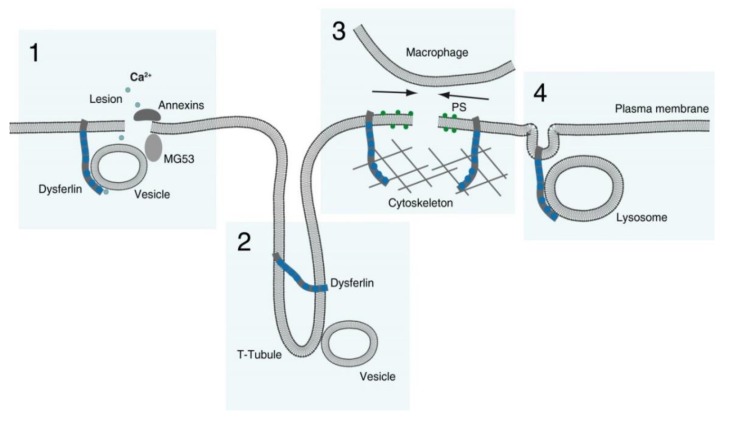
Mechanisms of dysferlin function in membrane repair. The model shows four possible and nonexclusive contributions of dysferlin to plasma membrane repair: (**1**) local formation of membranous patch or plug, triggered by calcium entry and supported, amongst others, by MG53 and annexins; (**2**) biogenesis and maintenance of the T-tubule system as a possible membrane reservoir; (**3**) cytoskeleton-dependent sorting of phosphatidylserine (PS) for the recruitment of macrophages, simultaneous contraction and subsequent sealing of the membrane wound and (**4**) exocytosis of lysosomes.

**Figure 5 cells-09-00534-f005:**
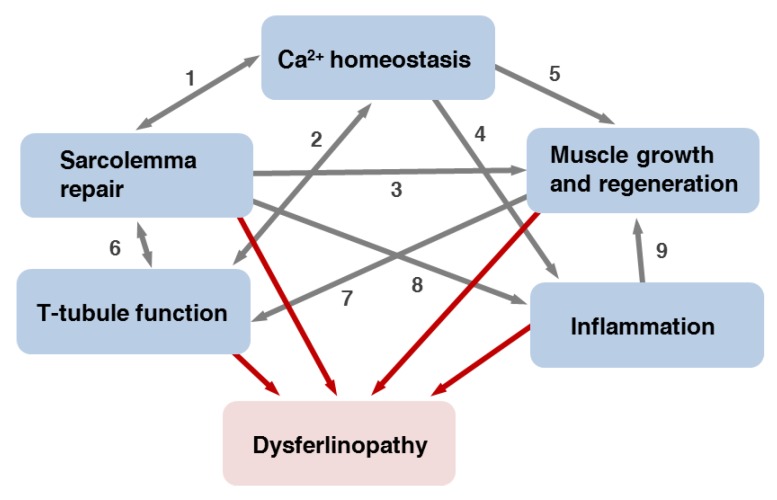
The network of dysferlin functions. Malfunctioning of one or several aspects contributes to the pathology of dysferlinopathies (red arrows). (1) Impaired sarcolemma repair leads to changes in Ca^2+^ homeostasis and in turn could be affected by intracellular Ca^2+^ compartmentalization and signaling. (2) The T-tubule system is necessary to maintain Ca^2+^ homeostasis and in turn could be affected by abnormalities in Ca^2+^ signaling. (3) Deficits in sarcolemma and T-tubule system repair may cause death of damaged myofibers and promote cycles of the muscle regeneration. Leakage of the muscle fibers contents may change properties of the regenerative niche. (4) Changes in Ca^2+^ compartmentalization and signaling in myofibers can result in dysregulation of, e.g., cytokines secretion and prolonged inflammatory responses. (5) Dysregulation of Ca^2+^ homeostasis may lead to myofibers death, which promotes cycles of muscle regeneration. (6) Sarcolemma repair may depend on the function of T-tubule system as a membrane reservoir and affect T-tubule system function via changes in Ca^2+^ homeostasis. (7) T-tubule system function may be affected by abnormalities in its structure arising during dysferlin-deficient muscle regeneration. (8) Malfunctioning of sarcolemma repair enhances leakage of damage-associated molecules, e.g., annexin A2, promoting inflammation. (9) Prolonged inflammation may result in incomplete cycles of regeneration and pro-inflammatory signaling may inhibit myogenesis.

**Table 1 cells-09-00534-t001:** Distribution of disease causing missense mutations in human dysferlin. The affected by missense mutations residues were identified in UMD-DYSFv1-4 dataset [179].

Domain	Amino acids ^1^	Mutations ^2^	% Mutations ^3^
C2A	101	3	3.0
C2A-C2B ICR	121	6	5.0
C2B	96	12	12.5
FerI	71	4	5.6
C2C	114	9	7.9
C2C-FerA ICR	200	7	3.5
FerA	65	1	1.5
FerB	74	4	5.4
DysF	227	21	9.3
C2D	108	2	1.9
C2D-C2E ICR	77	2	2.6
C2E	99	7	7.1
C2E-C2F ICR	142	7	4.9
C2F	99	6	6.1
C2F-C2G ICR	134	12	9.0
C2G	129	12	9.3
C2G-TM ICR	104	6	5.8
TM	23	1	4.4
Extracellular domain	15	2	13.3

^1^ Total number of amino acid residues in the domain. ^2^ Number of known amino acid residues found with missense mutations. ^3^ Percent of amino acid residues found with missense mutations. ^4^ Interconnecting region.

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
