# Peer review of "Functions of Vertebrate Ferlins"

_cells, 2020, doi:10.3390/cells9030534_

Round 1
Reviewer 1 Report
Bulankina et al. wrote an excellent review focused on the structures and functions of vertebrate ferlins. This review includes comprehensive information of six vertebrate ferlins from almost all previous paper and is well organized from their structural characters and physiological functions, to the association with diseases. This review will provide important clinical and scientific implications of ferlins, and be of broad interests to many scientists as well as clinicians, especially in the field of myology. However, I have some concerns on the manuscript.
The review is too long and sometimes there are repetitive and redundant descriptions. Especially Section 1 (Figure 1) should be omitted or be concise, because the authors discuss the functions of each ferlin in details later. The first paragraph in Section 12 contains, a long story Ca2+-dependent SNARE and Synaptotagmin dependent membrane fusion, should be put in Section 10, the second paragraph should be in Section 13. Section 15 is nothing to be said. Almost all are speculative and not give any direct relationship to dysferlin function and cancer formation.
Table 1 has no meaning. At line 630, "However, an analysis…are on average twofold more susceptible to this type of mutation than other region". This is not indicative of susceptibility of the disease causing, because they counted the number of mutated residues mutation, but not count the number of the patient with the mutations in each domain.
From Line 682, “Impaired muscle growth and regeneration” is also too speculative. As they mentioned, P21 is critical for juvenile to adult transition of satellite cells in the skeletal muscles in mice. However, almost murine muscular dystrophic models present severe dystrophic features at these days. Muscle damage would be due to locomotion and myofiber development, but not due to satellite cell properties.
Figure 5 should be improved. Some are related to normal muscle and inflammation is only related to disease status. What is a connection between Ca dishomeostasis and inflammation or defects in muscle growth and regeneration? Indicate direction (cause or result) would be helpful.
Reviewer 2 Report
This reviewer really appreciates the efforts made by the authors in this review.
This review on “Functions of vertebrate Ferlins” is complete and up to date. Different families of ferlins, structure and tissue expression are clearly developed in the review, using nice and well organised figures/tables. The Role of different ferlins in muscle is very clear. It successively describes the role of ferlins in membrane repair, in muscle growth and regeneration. It then goes in more details regarding the function of specific ferlins in muscle and non-muscle cells. It then describes the pathomechanisms of dysferlinopathies, deafness and cancer. It highlights the pitfalls of the current knowledge of the ferlin’s functions, and propose some hypothesis.
Overall, it is an excellent review.
This reviewer would only suggest to the authors to try to improve the figure 5. Here are few suggestions:
May be a “box” for “other function” is missing. May be some more info could be added in the boxes eg: “reduced sarcolemma repair”, etc. Maybe arrows and/or colour coding (green or red for activation or inhibition) in between some boxes could improve the reading of the graph.Author Response
Please see the attachment
